# Methoxyflurane for Relief of Procedural Pain in Burn Patients: A Prospective Single-Centre Evaluation Study

**Andreas Creutzburg** [1,*], **Martin R. Vestergaard** [1], **Pernille Pape** [1], **Caroline Hjelmdal** [1], **Filip Rangatchew** [2], **Rikke Holmgaard** [2] and **Lars S. Rasmussen** [1,3]

1   Department of Anesthesiology, Center of Head and Orthopedics, Rigshospitalet, Copenhagen University Hospital, Inge Lehmanns Vej 6, Section 6011, DK-2100 Copenhagen, Denmark
2   Department of Plastic Surgery and Burn Treatment, Centre of Head and Orthopaedics, Rigshospitalet, Copenhagen University Hospital, Inge Lehmanns Vej 7, Section 7035, DK-2100 Copenhagen, Denmark
3   Department of Clinical Medicine, Copenhagen University Hospital, DK-2100 Copenhagen, Denmark
*   Correspondence: andreas.creutzburg.01@regionh.dk; Tel.: +45-20-76-85-70

**Abstract:** *Background*: Procedural pain in burn patients continues to be a major problem. Frequently used analgesics, such as opioids, may have various side effects, including respiratory depression, nausea, and vomiting. Inhaled methoxyflurane has been used in the pre-hospital setting for trauma-related pain. This pilot study aimed to investigate the feasibility of using methoxyflurane for pain relief during dressings changes for burns in the hospital setting. *Methods*: In this investigator-initiated pilot study, we included burn patients undergoing dressing changes in the burn ward. The primary outcome was the maximal pain level experienced by the patient during the procedure on a verbal rating scale of 0 to 100. Furthermore, patient satisfaction and the nurse's assessment of the patient's pain were reported. We also reported the presence of nausea, vomiting, coughing, and headache, along with changes in the pulse rate, oxygen saturation, and arterial blood pressure. *Results*: We included 12 patients in the period of June 2021 to July 2022. The median patient-reported maximal procedural pain was 60 (interquartile range (IQR), 37–80), which corresponded well with the nurse's rating of a median of 57 (IQR 28–67). The patients were satisfied with methoxyflurane as an analgesic, with a median score of 96 (IQR 96–100). One patient reported coughing after the procedure, and another patient experienced nausea one week after the procedure. No clinically important haemodynamic changes during administration were detected. *Conclusions*: Methoxyflurane was found to be feasible for pain relief in burn patients undergoing dressing changes in the burn ward.

**Keywords:** pain management; burn patients; methoxyflurane

## 1. Introduction

Pain relief is a major challenge in the management of patients with severe burns. During hospitalisation, burn patients often need regular dressing changes as part of their care [1,2]. High pain levels in burn patients are associated with the development of chronic pain [3,4], a prolonged length of stay [1,3], and an increased level of anxiety [5,6]. Inadequate pain control is unfortunately not uncommon, yet optimal pain management remains unsolved [4]. Different pain management strategies have been attempted, including different opioids [2], ketamine [6], and lidocaine [7].

The side effects of some of these drugs are respiratory depression, tolerance development, nausea, and vomiting [2,4]. In addition, burn patients have important changes in physiology with the alteration in the total body water levels [4,8]. An optimal analgesic agent for burn patients should be rapid in onset and easy to administer, preferably without the need for intravenous access, and should have minimal side effects [5].

Inhaled methoxyflurane was first introduced by Packer et al. in the 1960s for general anaesthesia in Europe and the United States [9]. Methoxyflurane is no longer used for general anaesthesia due to dose-dependent acute renal failure and liver toxicity [3].

Methoxyflurane was reintroduced in Australia and New Zealand for acute pain management in trauma patients in the 1970s [10,11]. Using a self-administered inhaler, patients receive a low concentration of methoxyflurane for only a short period. In this case, there is a low risk of the serious side effects of methoxyflurane, which are seen after prolonged exposure [5,10]. To date, methoxyflurane has been used for more than 60 years [5,9], and previous studies have shown that methoxyflurane is safe to use in patients in general [10,12].

This study aimed to evaluate methoxyflurane's effect in burn patients and secondarily to assess side effects. The primary endpoint was the maximum pain level during the dressing change.

## 2. Materials and Methods

### 2.1. Study Design

This was an investigator-initiated, prospective, single-centre clinical evaluation study investigating the efficacy of methoxyflurane as an analgesic in burn patients undergoing dressing changes in the burn ward. All the enrolled patients received methoxyflurane during the dressing changes, with no control group.

### 2.2. Statement of Ethics

The study was conducted in accordance with the Helsinki declaration and approved by the regional Research Ethics Committee (H20081879) and the Danish Medicines Agency. Before the enrolment of the first patient, it was registered in a trial register (EudraCT number: 2020-005865-14). The data management was approved by the relevant authority (Pactius ID number: P-2021-325). All the patients gave written informed consent.

### 2.3. Eligibility Criteria

The inclusion criteria were: an adult ($\geq$18 years) capable of giving informed consent. The exclusion criteria were: pregnancy (a positive pregnancy test); known kidney injury, defined as a GFR below 60 mL/min/1.73 m [2]; known liver injury; previous liver damage due to halogenated inhalational anaesthetics; or genetic disposition to malignant hyperthermia. We also excluded patients who were being treated with isoniazid, gentamycin, tetracycline, colistin, or amphotericin B at the time of the study. Potentially eligible patients were screened by the burn surgeon, who contacted the investigators.

### 2.4. Study Site

This study was undertaken at Copenhagen University Hospital, Rigshospitalet, a tertiary 1200-bed hospital in central Copenhagen, Denmark. Our institution offers highly specialised treatment for burns in Denmark and is verified as a burn centre by the European Burn Association.

### 2.5. Intervention

Methoxyflurane (Penthrox®, Mundipharma A/S, Vedbæk, Denmark) was inhaled through a hand-held disposable vaporiser for self-administration (Figure 1). Just before initiating the dressing change, the vaporiser was loaded with 3 mL of methoxyflurane, which the patient then held throughout the entire procedure. Due to the drug's high lipid solubility, an analgesic effect was obtained within minutes after administration [9]. Methoxyflurane is understood to work by stimulating the GABA and glycine receptors in the central nervous system. It is not recommended to administer more than 2 ampules of methoxyflurane of 3 mL each per day, and one should not exceed 15 mL weekly. A total of 1 ampule of methoxyflurane normally provides pain relief for 20 to 30 min [5,13]. There was no need for preprocedural fasting [14]. On the day of administration, the patients were given their usual medication, including oral and intravenous analgesics. During the procedure, only the nursing staff performing the dressing change and research personnel were present.

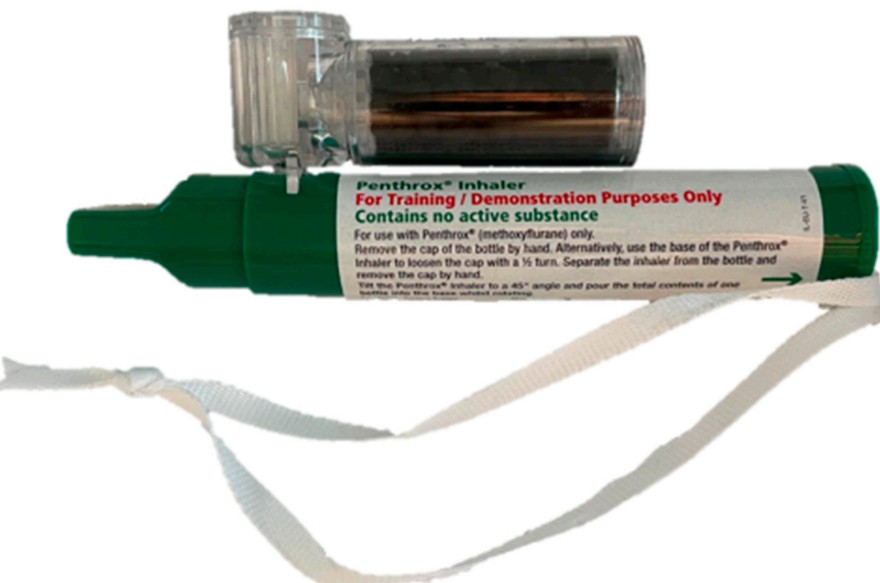

**Figure 1.** The Penthrox® inhaler with a carbon fibre chamber and the ampule of 3 mL of methoxyflurane. Through the opening on the top, the patients can increase the inspired concentration of methoxyflurane for short periods of time [3].

### 2.6. Data Collection

The data extracted from the patients' medical records included sex, age, height, weight, percentage of the total burned surface area, time since injury and last operation, the ASA classification, and information on the ongoing analgesic treatment.

Before and immediately after the procedure, patients rated their pain level on a verbal rating scale from 0 to 100 [15]. They also reported nausea and vomiting. We prospectively recorded the occurrence of speech difficulties, inebriation, drowsiness, headache, coughing, dry mouth, and taste disorder as adverse events. On a scale from 0 to 100 [15], patients were asked to rate their satisfaction. The nurses were asked to assess the maximal pain level of the patient during the procedure [15]. Furthermore, we recorded the duration of the procedure and whether patients moved in response to procedure-related pain. The patients were monitored with continuous pulse oximetry and blood pressure readings. The follow-up was conducted after seven days following administration to assess the safety and record any serious adverse events related to the administration of methoxyflurane.

Data were collected and managed using the Research Electronic Data Capture (RED-Cap) electronic data capture tools hosted by Rigshospitalet [16,17]. REDCap is a secure, web-based software platform designed to support data capture for research studies.

### 2.7. Statistical Analysis

Categorical variables were presented as frequencies (counts and proportions). Numerical variables were presented as medians with interquartile ranges (IQR). Only descriptive statistics were performed in this pilot study. We estimated that 30 patients would be needed to show that the true occurrence of any adverse event was lower than 12% with a 95% confidence interval if no event was recorded in that sample.

All the analyses were performed using R version 4.2.0 (R Project for Statistical Computing) [18] and RStudio version 2022.2.2.485 [19].

### 3. Results

The data were collected from the 25 June 2021 to the 31 July 2022. In this period, a total of 26 patients were screened for eligibility to participate in the study. Of these, 14 patients were excluded (Figure 2). The pilot study was terminated after 12 patients due to a low inclusion rate.

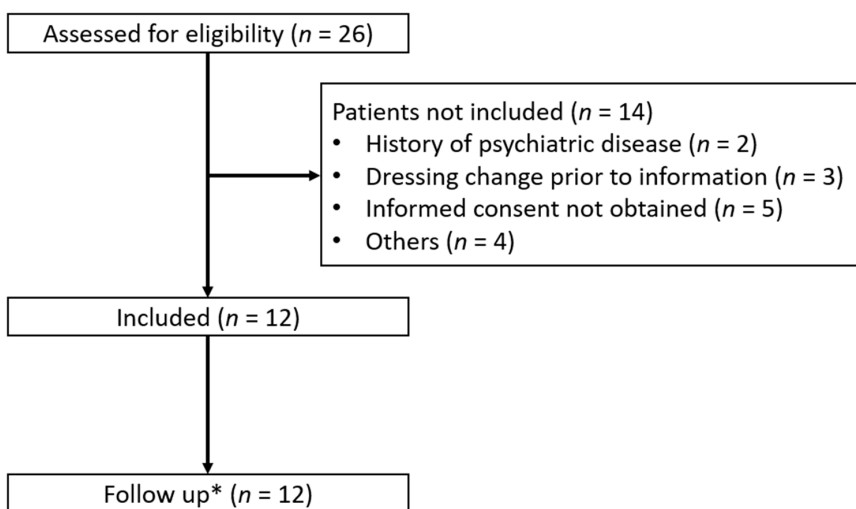

**Figure 2.** Screening and inclusion of the patients receiving dressing changes with methoxyflurane as analgesia. *: follow-up was performed seven days after the dressing change in the burn ward.

The median age was 56 years (50–68), and most burn patients were men (77%). The patients had a median BMI of 26.5 kg/m$^2$ (22.8–28.2) and were equally divided between ASA II and ASA III. The patients received methoxyflurane 22 days (17–34) after their injury and 6 (5–7) days after the latest operation.

The median total burned surface area was 10.5% (4.5–19.2), and the preprocedural pain level was 40 (18–50). The procedure could be completed for all the patients and lasted for 44 min (34–56). Most patients received a combination of oral analgesics (Table 1).

**Table 1.** Baseline characteristics of the study population receiving methoxyflurane for dressing changes in the burn ward *.

|  | *n* = 12 |
| --- | --- |
| **Patient characteristics** |  |
| Age, years | 56 (50–68) |
| Sex, male | 8 (67%) |
| BMI [#], kg/m$^2$ | 26.5 (22.8–28.2) |
| ASA classification |  |
|     ASA II | 7 (58%) |
|     ASA III | 5 (42%) |
| **Burn related characteristics** |  |
| Percentage of total burn surface area, % | 10.5 (4.5–19.2) |
| Days since burn injury, days | 22 (17–34) |
| Days since last operation, days | 6 (5–7) |
| Pain level before procedure [%] | 35 (18–50) |
| Procedure duration, minutes | 44 (34–56) |
| **Ongoing pain treatment** |  |
| Oral analgesics |  |
|     Paracetamol, 1 g × 2–4 | 12 (100%) |
|     Gabapentin, 300–600 mg × 2–3 | 10 (83%) |
|     Ibuprofen, 400 mg × 2–3 | 4 (33%) |
|     Morphine [¤] | 8 (67%) |
|     Others [¥] | 4 (33%) |
| Intravenous |  |
| Morphine [¤] | 2 (17) |

*: continuous variables are presented as medians with the first and third quartile and categorical variables as counts and percentages. [#]: body mass index. [%]: verbal rating scale. [¤]: average daily intake, 36 mg, intravenous morphine 20 mg as needed. [¥]: sertraline (100 mg × 1) and nortriptyline (10 mg × 1–2) and methadone 5 mg × 3.

We found that the maximal reported pain level during the dressing change procedure was 60 (37–80) (Table 2). The patient satisfaction score with methoxyflurane was 96 (69–100). The nurses' assessment of the maximal pain level during the procedure was a median score of 57 (26–66). Coughing was seen in one patient after the procedure (8%, 95% CI (1%;35%)). In the following week after the procedure, nausea was reported in one patient (8%, 95% CI (1%;35%)). No change was found in the haemodynamic parameters or oxygen saturation. No serious adverse events were recorded (Table 2).

**Table 2.** Primary and secondary outcomes of patients receiving methoxyflurane for dressing changes in the burn ward *.

|  |  | 95% CI |
| --- | --- | --- |
| **Primary outcome** |  |  |
| Maximal pain during the procedure [%] | 60 (37–80) |  |
| **Secondary outcomes** |  |  |
| Procedure related outcomes |  |  |
| Patient satisfaction [%] | 96 (69–100) |  |
| Nurse-assessed pain level [%] | 57 (2–6) |  |
| Patient movement | 1 (8%) |  |
| Serious adverse events and reactions | 0 (0%) | 0–24% |
| Adverse events |  |  |
| During the procedure [□] | 1 (8%) | 1–35% |
| 1 week after the procedure [$] | 1 (8%) | 1–35% |

|  | **Before** | **After** |
| --- | --- | --- |
| Physiologic changes |  |  |
| Oxygen saturation, % | 98 (97–99) | 99 (98–100) |
| Pulse, beats/min | 92 (72–99) | 89 (73–96) |
| MAP [§], mmHg | 106 (96–112) | 97 (87–109) |
| Lowest saturation during procedure, % | 96 (94–98) |  |

*: continuous variables are presented as medias with interquartile range and categorical variables as counts and percentages. [□]: includes nausea, vomiting, headache, and coughing. [$]: includes nausea, headache, and coughing. [§]: mean arterial blood pressure. [%]: verbal rating scale.

## 4. Discussion

In this study, we investigated whether methoxyflurane could be useful in facilitating dressing changes in burn patients. The maximum pain score was a median of 60 (37–80), which was consistent with nurses' assessment (57 (28–67)).

Our study had the important strength that the prospective study design resulted in the 100% completeness of data, adding reliability to our outcome assessments. The most important limitation of our study was that no control group was included, and therefore, we could not demonstrate the effectiveness compared to a standard treatment with other analgesic agents or a placebo. The study was performed without a control group because the aim was to investigate if it is possible to complete dressing changes before a possible implementation as the standard treatment.

Additionally, another limitation was the low inclusion rate. Originally, 30 patients were set to be enrolled, but after 13 months of active inclusion, only 26 patients were found to be eligible for inclusion, and only 12 patients were finally included in the study. The low inclusion rate was related to the strict exclusion criteria, especially concerning kidney and liver injury in this group of patients, who had typically been critically ill and treated in the intensive care unit. This decision was based on the careful consideration of safety because of the limited experience of the use of methoxyflurane in burn patients, even though the concentration would not exceed the recommended maximum dose under any circumstance [10,12]. Several patients refused participation in a scientific study, and others had a history of psychiatric disease, which did not allow for their inclusion or cooperation with the study procedures.

We found that the patients were very satisfied with methoxyflurane as an analgesic for dressing changes. One reason for this could be that the patients controlled the administration themselves and had the opportunity to increase the dosage of methoxyflurane. Side effects were uncommon and non-serious, as only one patient reported coughing after the procedure, and another reported a feeling of slight nausea during the week after the intervention. Patients with burns to the upper extremities presented a challenge regarding administration, as the patient had to use their arms to hold the inhaler. Patients maintained consciousness throughout the procedure without the need for intravenous access, which is a major advantage. Methoxyflurane has not been used widely in burn treatment before, but the nurses soon appreciated this new method and were well-pleased with the effect, as they were able to complete the dressing changes without supplemental analgesics, which may prolong the procedure and lead to discomfort for the patient because of the wait for analgesic onset. This was also the case for Borobia et al. Here, the need for supplemental analgesia was low in the acute pain management of patients treated with methoxyflurane [20]. We found a median procedure time of 44 min (34–56), enabling the dressing change to be completed with one or two dosages of methoxyflurane in most cases. Additionally, the nurses reported that they were able to complete the procedure faster than normal. Most patients only received oral analgesics as usual. Thus, this represents a novel approach, and it is our hope that some burn patients in the future will be able to undergo dressing changes in the burn ward itself instead of going to the operating room. Further, methoxyflurane could possibly reduce the personnel cost of dressing changes since the patients themselves are in control of the analgesic treatment during the procedure.

Recent studies of methoxyflurane primarily included patients undergoing a variety of minor procedures. Gaskell et al. reviewed the use of methoxyflurane for painful or uncomfortable procedures, including colonoscopies, biopsies, and dressing changes for burns. In a subset of patients, they found a median pain level of 2 among burn patients on a scale of 0–10 [21], with 10 being the worst pain score. Gaskell et al. possibly assessed the analgesic effect more fairly since incident pain cannot be avoided completely. Considering this, a median pain score of 60 in our study might seem unacceptable. However, the patients reported their maximal procedural pain. Along with a low pain level during the procedure, Packer et al. showed that patients were well-pleased with methoxyflurane during a dressing change, which was in agreement with the findings in this study [9]. Firn et al. reported that burn patients did not need supplemental opioids during dressing change, which was associated with a low incidence of nausea and vomiting, and they experienced the benefit of slight drowsiness, which facilitated the dressing change [22].

Nguyen et al. reported that during colonoscopies with the use of methoxyflurane for sedation, hypotension and tachyarrhythmia occurred in 1% of cases, while respiratory depression was not seen [23]. We did not find any clinically important change in the pulse rate, oxygen saturation, or arterial blood pressure during administration. The lowest oxygen saturation during the procedure was 96% (94–98), which is acceptable. Previous studies have found that oxygen saturation levels below 93% occurred in less than 1% of administrations [21].

In the prehospital setting, methoxyflurane has been shown to decrease pain scores by 40% to 50%. In this setting, the pain level fluctuates and is related to various interventions similar to those observed in the burn ward [24,25]. Based on the high patient satisfaction and the low rate of side effects, it could be reasonable to use methoxyflurane for procedural pain in burn patients, but it is desirable to conduct a randomised control trial where a benefit can be adequately assessed through a comparison with other drugs.

## 5. Conclusions

In conclusion, this study suggests that methoxyflurane can be used in patients with burns, with tolerable self-reported pain levels during dressing changes and patient satisfaction. No clinically important adverse events were seen among our 12 patients.

**Author Contributions:** Conceptualization, M.R.V., R.H. and L.S.R.; investigation, A.C., M.R.V., P.P. and C.H.; methodology, F.R., R.H. and L.S.R.; supervision, M.R.V. and L.S.R.; writing—original draft, A.C.; writing—review and editing, A.C., M.R.V., P.P., C.H., F.R., R.H. and L.S.R. All authors have read and agreed to the published version of the manuscript.

**Funding:** This research received no external funding.

**Institutional Review Board Statement:** The study was conducted in accordance with the Declaration of Helsinki and the regional Research Ethics Committee (H20081879).

**Informed Consent Statement:** Informed consent was obtained from all patients involved in the study.

**Data Availability Statement:** Not applicable.

**Conflicts of Interest:** The authors declare no conflict of interest.

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
