# Peer review of "Methoxyflurane for Relief of Procedural Pain in Burn Patients: A Prospective Single-Centre Evaluation Study"

_2673-1991, doi:10.3390/ebj3040047_

Round 1

Reviewer 1 Report

Overall, this was a well done and interesting proof-of-principle study the efficaciousness of methoxyflurane in burn pain analgesia. This method for analgesia is simple to execute and could have a significant impact for improving the patient experience during brief but very painful procedures.

One minor point could be expounded upon, just for additional clarification. It is unclear why a period of 7 days was chosen for follow-up? Was it expected that effects from a volatile anesthetic could manifest that long after the procedure? It would be useful to have a very brief stated rationale.  

Author Response

Point 1: Overall, this was a well-done and interesting proof-of-principle study on the efficaciousness of methoxyflurane in burn pain analgesia. This method for analgesia is simple to execute and could have a significant impact on improving the patient experience during brief but very painful procedures.

One minor point could be expounded upon, just for additional clarification. It is unclear why a period of 7 days was chosen for follow-up. Was it expected that effects from a volatile anaesthetic could manifest that long after the procedure? It would be useful to have a very brief stated rationale.  

Response 1: Thank you for the comment. The follow-up period of seven days was chosen to allow the detection of any adverse events after the administration of methoxyflurane which is known to be associated with liver and renal toxicity after prolonged exposure in high concentration.  

Reviewer 2 Report

Creutzburg et al conducted a study to investigate the feasibility of using methoxyflurane for pain relief during dressing change in patients with burn injuries. However, this study has limited novelty. The study also does not have a comparison arm, which is a significant limitation as the authors themselves pointed out. My comments are below:

Major points:

1. Methoxyflurane was found to be well-tolerated and effective during burn dressing changes very early in history (Laird and Chrystal, 1972 in Postgraduate Medical Journal). Wasiak et al in 2012 published in the International Wound Journal a case series of 15 burn patients using inhaled methoxyflurane for pain during burn wound care procedures including dressing changes and demonstrated that methoxyflurane was well tolerated and effective in this setting. In addition, Gaskell et al in 2016 published in the Journal Anaesthesia a randomized crossover study of burn patients undergoing burn-dressing procedures. They compared methoxyflurane and ketamine-midazolam and found that 5/8 patients preferred methoxyflurane, methoxyflurane had a better side effect profile and the self-administration of methoxyflurane was associated a sense of control. Given these prior findings already published in the literature, I am not sure how the results from the authors in this study would contribute to the existing body of literature.

2. The authors acknowledged the fact that their study does not have a comparison arm and this was a significant limitation of their study. The Gaskell et al 2016 study that I mentioned in my comment above, on the other hand, included ketamine-midazolam as a comparison arm. The authors should consider revising their study design and include a comparison arm – this would not only strengthen the rigor of their study design but also make their results more clinically relevant to help clinicians decide what analgesic agents to use in the setting of burn dressing change.

Minor points:

3. Line 124: “In thus period” should be revised to read “In this period”.

Author Response

Point 1: Methoxyflurane was found to be well-tolerated and effective during burn dressing changes very early in history (Laird and Chrystal, 1972 in Postgraduate Medical Journal). Wasiak et al in 2012 published in the International Wound Journal a case series of 15 burn patients using inhaled methoxyflurane for pain during burn wound care procedures including dressing changes and demonstrated that methoxyflurane was well tolerated and effective in this setting. In addition, Gaskell et al in 2016 published in the Journal Anaesthesia a randomized crossover study of burn patients undergoing burn-dressing procedures. They compared methoxyflurane and ketamine-midazolam and found that 5/8 patients preferred methoxyflurane, methoxyflurane had a better side effect profile and the self-administration of methoxyflurane was associated a sense of control. Given these prior findings already published in the literature, I am not sure how the results from the authors in this study would contribute to the existing body of literature.

Response 1: Thank you for the comments. Methoxyflurane has not previously been used in our institution. Therefore, it was necessary to investigate the feasibility of using methoxyflurane in the ward before potential implementation by assessing how well the patients and nursing staff would accept this new form of analgesia in the burn department. This is different from previous studies, where a doctor was present during the administration of methoxyflurane. It is our hope, that dressing changes in the future can be done without the need for a doctor present or that the management of painful procedures can be done in the burn department without the need of going to the operating room for the dressing change. Hereby, valuable resources could be reallocated. This has now been clarified in the manuscript in lines 186-189. It should also be noted that most of the mentioned studies included a heterogeneous group of patients and not only burn patients. Hereby, this study clarifies the usefulness of methoxyflurane for burn patients as a population due to their changed physiology as described in the introduction in the manuscript. Gaskell et al, state that the primary aim of their study is to investigate the feasibility, safety and efficacy in burn patients undergoing dressing changes, but they extended the inclusion criteria to also include other procedures. Hereby, we add further external validity to the finding regarding the burn patents in general. Finally, we also included the nurses’ assessment of the patient’s maximal pain during the procedure as an objective measurement of the maximal procedural pain. This was not done in any of the previous studies, and therefore adds to the literature.

Point 2: The authors acknowledged the fact that their study does not have a comparison arm and this was a significant limitation of their study. The Gaskell et al 2016 study that I mentioned in my comment above, on the other hand, included ketamine-midazolam as a comparison arm. The authors should consider revising their study design and include a comparison arm – this would not only strengthen the rigor of their study design but also make their results more clinically relevant to help clinicians decide what analgesic agents to use in the setting of burn dressing change.

Response 2: Thank you very much for this comment. We did consider a comparison arm for the study, however, in this feasibility study of methoxyflurane, we primarily aimed to investigate whether or not patients and nursing staff could complete dressing changes before a possible implementation of methoxyflurane as standard treatment for burn patients in our burn department with its inherited benefits.

Reviewer 3 Report

Major remarks:

Materials and Methods, Interventions

On the day of administration, patients were given their usual medication, including oral and intravenous analgesics.

-          How many patients received opioid analgesic agents and other analgetic drugs for chronic pain?

-          Was supplementary analgesia required in the patients with insufficient levels of analgesia? Which drugs were used for this purpose in patients with inadequate pain relief?

-          Did the patients encounter any  technical problem with the use of inhaler device?

The authors referred to ‘…Exclusion criteria were pregnancy (a positive pregnancy test); known kidney insufficiency, defined as a GFR below 60 ml/minute; known liver insufficiency…’

-          I would suggest replacing kidney and liver insufficiency with ‘kidney and hepatic injury’.

-          GFR is usually measured by using mL/min/1.73m2.

Were other nephrotoxic drugs (e.g., vancomysin, teicoplanin)  or contrast media used in these patients?

Results

The authors mentioned in the text that ‘…The median total burned surface area was 10.5% (4.5-19.2) and the preprocedural pain level was 40 (18-50)’.  However, in Table 1 they referred to that pain level before procedure was 35 [18-50]. How this discrepancy could be explained?

The authors stated in the result section:  ‘We found that the maximal reported pain level during the procedure of dressing 141 change was 60 (37-80).

-          In my opinion these results showed inadequate procedural pain treatment. I suggest discussing this finding in the discussion section.

-          I do not suggest including ‘feeling of slight nausea one week after the intervention’ in the catalogue of side effects.

Discussion section

-           In the discussion section the authors mentioned that ‘….The low inclusion rate was related to the strict exclusion criteria, especially with respect to kidney and liver insufficiency in this group of patients who had typically been critically ill and treated in the intensive care unit.

-           The authors referred to in the Results section that patients with mean TBSA of approximately 10% were included in this study (relatively small burns), I wonder how many of them were in critical condition after 22 days of stay in burn unit; how many of these patients had renal or hepatic failure? This point deserves further discussion.

-            

-           - I suggest mentioning the levels of consciousness (GCS)  after beginning of treatment.

Minor remarks:

Gaskell et al. might have assessed the 190 analgesic effect more fairly since incident pain cannot be avoided completely

-            Reference should be provided.

Conclusion section: high patient satisfactory

-           I would recommend replacing with ‘patients’ satisfaction’

Author Response

Point 1: Materials and Methods, Interventions

On the day of administration, patients were given their usual medication, including oral and intravenous analgesics.

-         How many patients received opioid analgesic agents and other analgetic drugs for chronic pain?

Response 1.1: Thank you very much for the thorough review of our manuscript.

We have now added information about ongoing analgesic medication for the included patients in Table 1.

-        Was supplementary analgesia required in the patients with insufficient levels of analgesia? Which drugs were used for this purpose in patients with inadequate pain relief?

Response 1.2: We only had one patient, who required supplemental analgesia during the dressing change. This was due to the procedure length, which was quite long. The patient received the recommended two doses of methoxyflurane, but since the dressing change was not completed when the effect of the methoxyflurane wore off after the second dose, the patient was given intravenous analgesia for the remaining part of the procedure. The patient reported, that he had been satisfied with the use of methoxyflurane during most of the procedure.

-          Did the patients encounter any  technical problem with the use of inhaler device?

Response 1.3: Some of our patients had burns on their upper extremities, and therefore, they had problems holding the vaporiser themselves. However, in all instances, patients still completed dressing changes with methoxyflurane as analgesia. The vaporiser was then held by another person who was directed by the patient.

The authors referred to ‘…Exclusion criteria were pregnancy (a positive pregnancy test); known kidney insufficiency, defined as a GFR below 60 ml/minute; known liver insufficiency…’

-          I would suggest replacing kidney and liver insufficiency with ‘kidney and hepatic injury’.

Response 1.4: Thank you. This has been implemented.

-          GFR is usually measured by using mL/min/1.73m2.

Response 1.5: This has been corrected.

-   Were other nephrotoxic drugs (e.g., vancomysin, teicoplanin)  or contrast media used in these patients?

Response 1.6: We scrutinized the patient’s medication list prior to inclusion because several drugs contraindicate the use of methoxyflurane. None of them received the drugs mentioned by the reviewer.

Point 2: Results

The authors mentioned in the text that ‘…The median total burned surface area was 10.5% (4.5-19.2) and the preprocedural pain level was 40 (18-50)’.  However, in Table 1 they referred to that pain level before procedure was 35 [18-50]. How this discrepancy could be explained?

Response 2.1: Thank you for pointing out this. This was a typing error, and we apologise. This has now been corrected to “preprocedural pain level: 35 [18-50]” in Table 1.

The authors stated in the result section:  ‘We found that the maximal reported pain level during the procedure of dressing 141 change was 60 (37-80).

-          In my opinion these results showed inadequate procedural pain treatment. I suggest discussing this finding in the discussion section.

Response 2.2: We agree that a pain level of 60 on a scale of 0 to 100 may seem unacceptable but it should be taken into account that patients were asked to report their maximal pain level during the procedure. That may not be a fair representation of the effect of methoxyflurane for the entire procedure. This is now addressed in the discussion.

-          I do not suggest including ‘feeling of slight nausea one week after the intervention’ in the catalogue of side effects.

Response 2: This side effect was included because it is one of the most common. Previous studies have found that 1-10% of patients experience nausea due to methoxyflurane.

Point 3: Discussion section

-         In the discussion section the authors mentioned that ‘….The low inclusion rate was related to the strict exclusion criteria, especially with respect to kidney and liver insufficiency in this group of patients who had typically been critically ill and treated in the intensive care unit.

-         The authors referred to in the Results section that patients with mean TBSA of approximately 10% were included in this study (relatively small burns), I wonder how many of them were in critical condition after 22 days of stay in burn unit; how many of these patients had renal or hepatic failure? This point deserves further discussion.

Response 3.1: Thank you for this highly relevant comment. The low inclusion rate was related to the strict exclusion criteria that we used for safety reasons. The most sick patients were therefore not studied and as a consequence, our study group consisted of those with relatively small burns.

-          I suggest mentioning the levels of consciousness (GCS)  after beginning of treatment.

Response 3.2: We carefully assessed the level of consciousness during the procedure but the Glasgow Coma Scale is not useful during dressing changes. All patients remained conscious throughout the procedure

Point 4: Minor remarks:

Gaskell et al. might have assessed the 190 analgesic effect more fairly since incident pain cannot be avoided completely

-            Reference should be provided.

Response 4.1: We refer to “Gaskell et al: Self-administered methoxyflurane analgesia” in the discussion section.

Conclusion section: high patient satisfactory

-           I would recommend replacing with ‘patients’ satisfaction’

Response 4.2: Thank you. This has been implemented.

Reviewer 4 Report

Thank you for giving me the opportunity for reviewing this interesting article on the use of methoxyflurane for the relief of peak procedural pain in patients with burns. I have some suggestions and questions, please see below.

Introduction

Could you please add some information regarding the working mechanism of methoxyflurane?

Material and methods

Line 83: I should state that the use of methoxyflurane is additional medication to the standard medication.

Line 84: Did all patients inhale the same quantity, in relation to body mass index? Or was there much difference?

Line 93: What do the authors mean by the usual medication? And were the doses changed in advance or were the usual doses administered?

Line 102: How many wound care procedures per patient using methoxyflurane have been carried out?

Line 106: Nurses were asked to assess pain levels. The first choice for pain measurement is always the patients’ self-report. Pain behaviour observation is only appropriate in patients unable to provide self-reports. When nurses use a global rating scale like the verbal numerical scale, to report the patients’ pain, it turns out to be unreliable. Please consider the references below. Could the authors explain why they choose for this manner of pain measurement?

De Jong et al. Reliability and validity of the pain observation scale for young children and the visual analogue scale in children with burns. Burns 31 (2005) 198-204.

De Jong et al. Reliability, validity and clinical utility of three types of pain behavioural observation scales for young children with burns aged 0-5 years. PAIN 150 (2010) 561–567.

Van der Does AJW. Patients’ and nurses’ ratings of pain and anxiety during burn wound care. Pain 1989;39:95–101.

Choinière M, Melzack R, Girard N, Rondeau J, Paquin MJ. Comparisons between patients’ and nurses’ assessment of pain and medication efficacy in severe burn injuries. Pain 1990; 40: 143-52.

Geisser ME, Bingham HG, Robinson ME. Pain and anxiety during burn dressing changes: concordance between patients’ and nurses’ ratings and relation to medication administration and patient variables. J Burn Care Rehabil 1995;16:165–71.

Line 106: Who was responsible for the datacollection of pain scores? Was it performed by the same nurse who rated the patients’ pain? If yes, what was the order of data collection: the patient self-report first, or the nurse report?

Line 109: Please accentuate that pulse rate and blood pressure are used as indicators for side effects of methoxyflurane and not as indicators for pain.

Results

Line 127: I would suggest pilot instead of trial.

Discussion

Line 176: In line 93 the researchers state that  usual medication consists of intravenous analgesics, but that was not necessary in this specific sample?

Since this is a feasibility study, were patients able to hold the inhaler during the entire procedure, or did they inhale now and then?

Author Response

Point 1: Introduction

Could you please add some information regarding the working mechanism of methoxyflurane?

Response 1: Thank you for the suggestion. We have added that the suggested mechanism is mediated through GABA and glycine receptors.

Point 2: Material and methods

Line 83: I should state that the use of methoxyflurane is additional medication to the standard medication.

Response 2.1: Thank you for the relevant suggestion. We have added information about ongoing pain therapy in these patients.

Line 84: Did all patients inhale the same quantity, in relation to body mass index? Or was there much difference?

Response 2.2: All patients received at least 1 ampoule of methoxyflurane, 3 ml. If the dressing change was not completed after 25 to 30 minutes, a second ampoule was prepared for the patient for the remaining time of the dressing change. The dosage is not dependent on the BMI.

Line 93: What do the authors mean by the usual medication? And were the doses changed in advance or were the usual doses administered?

Response 2.3: Please see our response 2.1

Line 102: How many wound care procedures per patient using methoxyflurane have been carried out?

Response 2.4: Each patient went through one dressing change using methoxyflurane. If patients need more than one dressing change, conventional analgesic agents were used for procedural pain. This was done as a precaution, so patients would not reach the maximum amount of methoxyflurane of 15 ml on a weekly basis.

Line 106: Nurses were asked to assess pain levels. The first choice for pain measurement is always the patients’ self-report. Pain behaviour observation is only appropriate in patients unable to provide self-reports. When nurses use a global rating scale like the verbal numerical scale, to report the patients’ pain, it turns out to be unreliable. Please consider the references below. Could the authors explain why they choose for this manner of pain measurement?

Response 2.5: Thanks for this comment. In our study, patients were asked to rate their maximum pain level during the procedure. After the procedure, the nurse doing the dressing change was asked to rate the patient’s maximal pain during the change in their view. The reason for choosing this was, that it would be comparable with the patients’ self-reported measurement of maximal pain during the procedure.

Line 109: Please accentuate that pulse rate and blood pressure are used as indicators for side effects of methoxyflurane and not as indicators for pain.

Response 2.6: We monitored blood pressure and oxygen saturation because some common side effects of methoxyflurane are hypoxia due to respiratory depression and hypotension due to a vasodilatory effect. The two measurements were not used to monitor pain levels during the procedure.

Point 3: Results

Line 127: I would suggest pilot instead of trial.

Response 3.1: Thank you for the suggestion. This has been included in the manuscript.

Point 4: Discussion

Line 176: In line 93 the researchers state that  usual medication consists of intravenous analgesics, but that was not necessary in this specific sample?

Response 4.1: During their admission, patients were given a standard analgesic regiment consisting of both oral and intravenous analgesic agents. On the day of administration of methoxyflurane, patients were not given additional drugs since they received methoxyflurane.

Since this is a feasibility study, were patients able to hold the inhaler during the entire procedure, or did they inhale now and then?

Response 4.2: Some of our patients had burns on their upper extremities, and therefore, they had problems holding the vaporiser themselves. However, in all instances, patients still completed dressing changes with methoxyflurane as analgesia. The vaporiser was then held by another person who was directed by the patient.

Round 2

Reviewer 2 Report

The authors did not adequately address my comments. I find some of the statements made by the authors in their responses to be unconvincing. 

1. The authors state in their response #1 "This is different from previous studies, where a doctor was present during the administration of methoxyflurane". This is simply a false statement. Gaskell et al 2016 reported on self-administered methoxyflurane where patients themselves titrate anesthesia to the desired level with supervision from nursing staff without needing a doctor or using the operating theatre. Wasiak et al 2014 also studied self-administered methoxyflurane and only in burn patients undergoing dressing changes. Therefore, the authors still have not demonstrated the novelty of their study in their response to my first comment. In addition, when revising for the next round of review, the authors should make sure to include in their manuscript how their study is novel, rather than just responding to my comments here.

2. Lacking a control group is a significant limitation and as such it should be clearly acknowledged in the manuscript.

3. Please use generic drug names and avoid using brand names such as ibumetin (ibuprofen is the generic name) in table 1 and noritren (Nortriptyline is the generic name). 

Reviewer 3 Report

 The authors have made appropriate changes and have improved the quality of the manuscript. 

Author Response

Thanks for the revisions.

Round 3

Reviewer 2 Report

The authors have responded to my prior comments. 

Please make sure all typos and grammatical errors are corrected.